# GRAPH NEIGHBORHOOD ATTENTIVE POOLING

## ABSTRACT

Network representation learning (NRL) is a powerful technique for learning low-dimensional vector representation of high-dimensional and sparse graphs. Most studies explore the structure and metadata associated with the graph using random walks and employ an unsupervised or semi-supervised learning schemes. Learning in these methods is context-free, because only a single representation per node is learned. Recently studies have argued on the sufficiency of a single representation and proposed a context-sensitive approach that proved to be highly effective in applications such as link prediction and ranking.

However, most of these methods rely on additional textual features that require RNNs or CNNs to capture high-level features or rely on a community detection algorithm to identify multiple contexts of a node.

In this study, without requiring additional features nor a community detection algorithm, we propose a novel context-sensitive algorithm called GAP that learns to attend on different parts of a node's neighborhood using attentive pooling networks. We show the efficacy of GAP using three real-world datasets on link prediction and node clustering tasks and compare it against 10 popular and state-of-the-art (SOTA) baselines. GAP consistently outperforms them and achieves up to $\approx 9\%$ and $\approx 20\%$ gain over the best performing methods on link prediction and clustering tasks, respectively.

## 1 INTRODUCTION

NRL is a powerful technique to learn representation of a graph. Such a representation gracefully lends itself to a wide variety of network analysis tasks, such as link prediction, node clustering, node classification, recommendation, and so forth.

In most studies, the learning is done in a context-free fashion. That is, the representation of a node characterizes just a single aspect of the node, for instance, the local or global neighborhood of a node. Recently, a complementary line of research has questioned the sufficiency of single representations and considered a context-sensitive approach. Given a node, this approach projects it to different points in a space depending on other contexts it is coupled with. A context node can be sampled from a neighborhood (Tu et al., 2017; Zhang et al., 2018), random walk (Ying et al., 2018), and so on. In this study we sample from a node neighborhood (nodes connected by an edge). Thus, in the learning process of our approach a source node's representation changes depending on the target (context) node it is accompanied by. Studies have shown that context-sensitive approaches significantly outperform previous context-free SOTA methods in link-prediction task. A related notion (Peters et al., 2018; Devlin et al., 2018) in NLP has significantly improved SOTA across several NLP tasks.

In this paper we propose GAP (**G**raph neighborhood **a**ttentive **p**ooling), which is inspired by *attentive pooling networks* (APN) (dos Santos et al., 2016), originally proposed for solving the problem of pair ranking in NLP. For instance, given a question $q$, and a set of answers $A = \{a_1, \ldots, a_z\}$, an APN can be trained to rank the answers in $A$ with respect to $q$ by using a two-way attention mechanism. APN is based on the prevalent deep learning formula for SOTA NLP, that is, *embed*, *encode*, *attend*, *predict* (Honnibal, 2018). Given a question-answer pair $(q, a_i)$, the APN model first projects the embedding of the pairs using two separate encoders, and the encoder can be a CNN or LSTM. The projection helps to capture n-gram context information and/or long-term dependencies in the input sequence. Next, a soft-alignment matrix is obtained by a mutual-attention mechanism that transforms these projections using a parameter matrix. Attention vectors are then computed through

a column-wise and row-wise pooling operations on the alignment matrix. Finally, the weighted sum of each of the above projections by its respective attention vector is computed to obtain the representations of the question and answer. Each candidate answer $a_i$ is then ranked according to its similarity with the question $q$ computed using the representations of $q$ and $a_i$.

Recently, APN have been applied to context-sensitive NRL by Tu et al. (2017), and the inputs are textual information attached with a pair of incident nodes of edges in a graph. Such information, however, has the added overhead of encoding textual information.

Though we adopt APN in GAP, we capitalize on the graph neighborhood of nodes to avoid the need for textual documents without compromising the quality of the learned representations. Our hypothesis is that one can learn high-quality context-sensitive node representations just by mutually attending to the graph neighborhood of a node and its context node. To achieve this, we naturally assume that the order of nodes in the graph neighborhood of a node is arbitrary. Moreover, we exploit this assumption to simplify the APN model by removing the expensive encode phase.

Akin to textual features in APN, GAP simply uses graph neighborhood of nodes. That is, for every node in the graph we define a graph neighborhood function to build a fixed size neighborhood sequence, which specifies the input of GAP. In the APN model, the *encoder* phase is usually required to capture high-level features such as n-grams, and long term and order dependencies in textual inputs. As we have no textual features and due to our assumption that there is no ordering in the graph neighborhood of nodes, we can effectively strip off the encoder. The encoder is the expensive part of APN as it involves a RNN or CNN, and hence GAP can be trained faster than APN.

This simple yet empirically fruitful modification of the APN model enables GAP to achieve SOTA performance on link prediction and node clustering tasks using three real world datasets. Furthermore, we have empirically shown that GAP is more than 2 times faster than an APN like NRL algorithm based on text input. In addition, the simplification in GAP does not introduce new hyperparameters other than the usual ones, such as the learning rate and sequence length in APN.

## 2 APN ARCHITECTURE

For the sake of being self-contained, here we briefly describe the original APN architecture. We are given a pair of natural language texts $(x, y)$ as input, where $x = (w_1, \ldots w_X)$ and $y = (w_1, \ldots, w_Y)$ are a sequence of words of variable lengths, and each word $w_i$ is a sample from a vocabulary $\mathbb{V}$, $X = |x|$, $Y = |y|$, and $X$ and $Y$ could be different. The APN's forward execution is shown in Fig. 1(A) and in the following we describe each component.

**Embed:** First embedding matrices of $x$ and $y$ are constructed through a lookup operation on an embedding matrix $\boldsymbol{E} \in \mathbb{R}^{d \times |\mathbb{V}|}$ of words, where $d$ is the embedding dimension. That is, for both $x$ and $y$, respectively, embedding matrices $\boldsymbol{X} = [\boldsymbol{w}_1, \ldots, \boldsymbol{w}_X]$ and $\boldsymbol{Y} = [\boldsymbol{w}_1, \ldots, \boldsymbol{w}_Y]$ are constructed by concatenating embeddings $\boldsymbol{w}_i$ of each word $w_i$ in $x$ and $y$, the *Embed* box in Fig. 1(A).

**Encode:** Each embedding matrix is then projected using a CNN or BI-LSTM encoder to capture inherent high-level features, the *Encode* box in Fig. 1(A). More formally, the embedded texts $\boldsymbol{X} \in \mathbb{R}^{d \times X}$, $\boldsymbol{Y} \in \mathbb{R}^{d \times Y}$ are projected as $\boldsymbol{X}_{enc} = f(\boldsymbol{X}, \Theta)$ and $\boldsymbol{Y}_{enc} = f(\boldsymbol{Y}, \Theta)$ where $f$ is the encoder, CNN or BI-LSTM, $\Theta$ is the set of parameters of the encoder, and $\boldsymbol{X}_{enc} \in \mathbb{R}^{c \times X}$ and $\boldsymbol{Y}_{enc} \in \mathbb{R}^{c \times Y}$, where $c$ is the number of filters or hidden features of the CNN and BI-LSTM, respectively.

**Attend:** In the third step, a parameter matrix $\boldsymbol{P} \in \mathbb{R}^{c \times c}$ is introduced so as to learn a similarity or soft-alignment matrix $\boldsymbol{A} \in \mathbb{R}^{X \times Y}$ between the sequence projections $\boldsymbol{X}_{enc}$ and $\boldsymbol{Y}_{enc}$ as:

$$\boldsymbol{A} = \tanh(\boldsymbol{X}_{enc}^T \boldsymbol{P} \boldsymbol{Y}_{enc})$$

Then unnormalized attention weight vectors $\boldsymbol{x}' \in \mathbb{R}^X$ and $\boldsymbol{y}' \in \mathbb{R}^Y$ are obtained through a column-wise and row-wise max-pooling operations on $\boldsymbol{A}$, respectively as $\boldsymbol{x}'_i = \max(\boldsymbol{A}_{i,:}), \boldsymbol{y}'_j = \max(\boldsymbol{A}_{:,j})$, where $0 \leq i < X$, $0 \leq j < Y$ and $\boldsymbol{A}_{i,:}$ and $\boldsymbol{A}_{:,j}$ are the $i$-th and $j$-th row and column of $\boldsymbol{A}$, respectively. Next, the attention vectors are normalized using softmax, $\boldsymbol{x} = \texttt{softmax}(\boldsymbol{x}')$

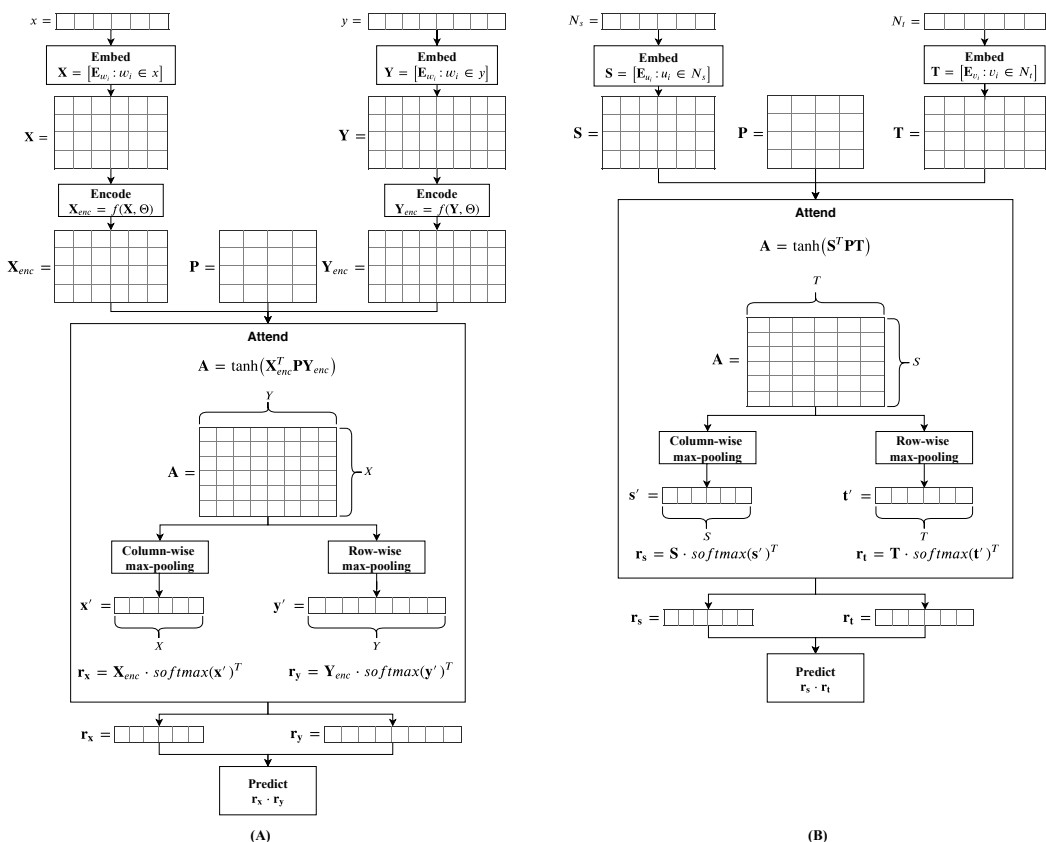

Figure 1: The APN (A) and GAP (B) models forward execution for question answering and context-sensitive node representation, respectively.

and $\boldsymbol{y} = \texttt{softmax}(\boldsymbol{y}')$. Finally, the normalized attention vectors are used to compute the final representations as $\boldsymbol{r}_x = \boldsymbol{X}_{enc} \cdot \boldsymbol{x}^T$ and $\boldsymbol{r}_y = \boldsymbol{Y}_{enc} \cdot \boldsymbol{y}^T$.

**Predict:** In the last step, the representations $\boldsymbol{r}_x$ and $\boldsymbol{r}_y$ will be used for ranking depending on the task on hand. For instance, in a question and answer setting, each candidate answer's representation $\boldsymbol{r}_y$ will be ranked based on its similarity score with the question's representation $\boldsymbol{r}_x$.

## 3 GAP

GAP adopts the APN model for learning the representations of node pairs in a graph $G = (V, E)$ with a set of $n$ nodes $V$ and $m$ edges $E$. $G$ can be a directed or undirected and weighted or unweighted graph. Without loss of generality we assume that $G$ is an unweighted directed graph.

We define a neighborhood function $N : V \to 2^V$, which maps each node $u \in V$ to a set of nodes $N_u \subseteq V$. A simple way of materializing $N_u$ is to consider the first-order neighbors of $u$, that is, $N_u = [v : (u, v) \in E \vee (v, u) \in E]$. An important assumption that we have on $N_u$ is that the ordering of the nodes in $N_u$ is not important. GAP capitalizes on this assumption to simplify the APN model and achieve SOTA performance. Even though one can explore more sophisticated neighborhood functions, in this study we simply consider the first order neighborhood.

Our goal is to learn node representations using the simplified APN based on the node neighborhood function $N$. Hence, akin to the input text pairs in APN, we consider a pair of neighborhood sequences $N_s = (u_1, \ldots, u_S)$ and $N_t = (v_1, \ldots, v_T)$ associated with a pair $(s, t) \in E$ of nodes $s$ and $t$, and $S = |N_s|$ and $T = |N_t|$. Without loss of generality we consider $S = T$. Recall that we assume the order of nodes in $N_s$ and $N_t$ is arbitrary.

| Dataset | #Nodes | #Edges | Features |
|---------|--------|--------|----------------|
| Cora    | 2277   | 5214   | Paper Abstract |
| Zhihu   | 10000  | 43894  | User post      |
| Email   | 1005   | 25571  | NA             |

Table 1: Summary of datasets, the *Features* column is relevant to some of the baselines not GAP

Given a source node $s$, we seek to learn multiple context-sensitive embeddings of $s$ with respect to a target node $t$ it is paired with. In principle one can learn using all pairs of nodes, however that is not scalable, and hence we restrict learning between pairs in $E$.

GAP's forward execution model is shown in Fig 1(B), and learning starts by embedding $N_s$ and $N_t$, respectively, as $\boldsymbol{S} = (\boldsymbol{u}_1, \ldots, \boldsymbol{u}_S)$ and $\boldsymbol{T} = (\boldsymbol{v}_1, \ldots, \boldsymbol{v}_T)$. Since there is no order dependency between the nodes in $N_s$ or $N_t$, besides being a neighbor of the respective node, we leave out the CNN or BI-LSTM based projections of $\boldsymbol{S}$ and $\boldsymbol{T}$ that could capture the dependencies. *No encoder!*

Thus, the next step of GAP is mutually attending on the embeddings, $\boldsymbol{S}$ and $\boldsymbol{T}$, of the graph neighborhood of the node pairs; the *Attend* box of 1(B). That is, we employ the trainable parameter matrix $\boldsymbol{P} \in \mathbb{R}^{d \times d}$ and compute the soft-alignment matrix, $\boldsymbol{A}$, between the neighbors of $s$ and $t$.

$$\boldsymbol{A} = \tanh(\boldsymbol{S}^T \boldsymbol{P} \boldsymbol{T}) \tag{1}$$

Here $\boldsymbol{A}$ is a soft-alignment matrix between every pair of nodes, $(u, v) \in N_s \times N_t$. Therefore, for each axis of $\boldsymbol{A}$, we proceed by pooling the maximum alignment score for each node to obtain the unnormalized attention vectors $\boldsymbol{s}'_i = \max(\boldsymbol{A}_{i,:})$ and $\boldsymbol{t}'_j = \max(\boldsymbol{A}_{:,j})$. As a result of the pooling operations, each neighbor $u \in N_s$ of the source node, $s$, selects a neighbor $v \in N_t$ of the target node, $t$, with the maximum alignment or similarity score. A similar selection is done for $v \in N_t$. This enables the source and target neighborhood sequences of the pair to influence each other in order to learn a context-sensitive representation of $s$ and $t$. The normalized attention vectors are then obtained by applying softmax as $\boldsymbol{s} = \texttt{softmax}(\boldsymbol{s}')$ and $\boldsymbol{t} = \texttt{softmax}(\boldsymbol{t}')$. Ultimately, we compute the context-sensitive representations $\boldsymbol{r}_s$ and $\boldsymbol{r}_t$ of the source and target nodes $s$ and $t$, respectively as $\boldsymbol{r}_s = \boldsymbol{S} \cdot \boldsymbol{s}^T$ and $\boldsymbol{r}_t = \boldsymbol{T} \cdot \boldsymbol{t}^T$.

**Optimization:** The objective of GAP is to maximize the likelihood of the graph (edges) by predicting edge similarities using the dot product of the source and target representations as $\boldsymbol{r}_s \cdot \boldsymbol{r}_t$; the *Predict* box of Fig 1(B). Hence, we employ a hard-margin loss given in Eq. 2.

$$\mathcal{L}(s, t) = \max(0, 1 - \boldsymbol{r}_s \cdot \boldsymbol{r}_t + \boldsymbol{r}_s \cdot \boldsymbol{r}_t^-) \tag{2}$$

where $\boldsymbol{r}_t^-$ is the representation of a negative target node $t^-$, that is $(s, t^-) \notin E$. The goal is to learn, in an unsupervised fashion, a context-sensitive embedding of nodes that enable us to rank the positive edges $(s, t) \in E$ higher than the negative pairs $(s, t^-)$.

Finally a word on the computational complexity of GAP that is proportional to the the number of edges, as we are considering each edge as an input pair.

## 4 EXPERIMENTAL EVALUATION

In this section we provide an empirical evaluation of GAP. To this end, experiments are carried out using the following datasets, and a basic summary is given in Table 1.

1. Cora Tu et al. (2017); Zhang et al. (2018): is a citation network dataset, where a node represents a paper and an edge $(u, v) \in E$ represents that paper $u$ has cited paper $v$.

2. Zhihu Tu et al. (2017); Zhang et al. (2018): is the biggest social network for Q&A and it is based in China. Nodes are the users and the edges are follower relations between the users.

3. Email Leskovec et al. (2007): is an email communication network between the largest European research institutes. A node represents a person and an edge $(u, v) \in G$ denotes that person $u$ has sent an email to $v$.

| Dataset | #Neighborhood ($S$ and $T$) | Dropout | Learning rate | Representation size |
|---------|------------------------------|---------|---------------|---------------------|
| Cora    | 100                          | 0.5     | 0.0001        | 200                 |
| Zhihu   | 250                          | 0.65    | 0.0001        | 200                 |
| Email   | 100                          | 0.8     | 0.0001        | 200                 |

Table 2: Conifguration of GAP

The first two datasets have features (documents) associated to nodes. For Cora, abstract of papers and Zhihu user posts. Some of the baselines, discussed beneath, require textual information, and hence they consume the aforementioned features. The Email dataset has ground-truth community assignment for nodes based on a person's affiliation to one of the 42 departments.

We compare our method against the following 10 popular and SOTA baselines grouped as:

- *Structure based methods*: DEEPWALK Perozzi et al. (2014), NODE2VEC Grover & Leskovec (2016), WALKLETS Perozzi et al. (2016), ATTENTIVEWALK Abu-El-Haija et al. (2017), LINE Tang et al. (2015):

- *Structure and attribute based methods*: TRIDNR Pan et al. (2016), TADW Yang et al. (2015), CENE Sun et al. (2016)

- Context-sensitive: CANE Tu et al. (2017), DMTE Zhang et al. (2018)

We are aware of another SOTA context-sensitive algorithm (Epasto & Perozzi, 2019) that do not require textual information, however we have not included it in our experiments because the source code is not available.[1] Now we report the experimental results carried out in two tasks, which are link prediction and node clustering. All experiments are performed using a 24-Core CPU and 125GB RAM Ubuntu 18.04 machine.

## 4.1 LINK PREDICTION

Link prediction is an important task that graph embedding algorithms are applied to. Particularly context-sensitive embedding techniques have proved to be well suited for this task. Similar to existing studies we perform this experiment using a fraction of the edges as a training set. We hold out the remaining fraction of the edges from the training phase and we will only reveal them during the test phase, results are reported using this set. All hyper-parameter tuning is performed by taking a small fraction of the training set as a validation set.

**Setup:** In-line with existing techniques (Tu et al., 2017; Zhang et al., 2018), the percentage of training edges ranges from 15% to 95% by a step of 10. The hyper-parameters of all algorithms are tuned using random-search. For some of the baselines, our results are consistent with what is reported in previous studies, and hence for Cora and Zhihu we simply report these results.

Except the "unavoidable" hyper-parameters (eg. learning rate, regularization/dropout rate) that are common in all the algorithms, our model has just one hyper-parameter which is the neighborhood sequence length (#Neighborhood $= S = T$), for nodes with smaller neighborhood size we use zero padding. As we shall verify later, GAP is not significantly affected by the choice of this parameter.

The quality of the prediction task is measured using the AUC score. AUC indicates the probability that a randomly selected pair $(u, w) \notin E$ will have a higher similarity score than an edge $(u, v) \in E$. Similarity between a pair of nodes is computed as the dot product of their representation. For all the algorithms the representation size $-d$ is 200 and GAP's configuration is shown in Table 2.

**Results:** The results of the empirical evaluations on the Cora, Zhihu, and Email datasets are reported in Tables 3, 4, and 5. GAP outperforms the SOTA baselines in all cases for Zhihu and Email, and in almost all cases for Cora. One can see that as we increase the percentage of training edges, performance increases for all the algorithms. As indicated by the "Gain" row, GAP achieves up to 9% improvement over SOTA context-sensitive techniques. Notably the gain is pronounced for

---

[1]We have contacted the authors and have been told that the code is not ready for release

| Algorithm | % of training edges | | | | | | | | |
|---|---|---|---|---|---|---|---|---|---|
| | 15% | 25% | 35% | 45% | 55% | 65% | 75% | 85% | 95% |
| DEEPWALK | 56.0 | 63.0 | 70.2 | 75.5 | 80.1 | 85.2 | 85.3 | 87.8 | 90.3 |
| LINE | 55.0 | 58.6 | 66.4 | 73.0 | 77.6 | 82.8 | 85.6 | 88.4 | 89.3 |
| NODE2VEC | 55.9 | 62.4 | 66.1 | 75.0 | 78.7 | 81.6 | 85.9 | 87.3 | 88.2 |
| WALKLETS | 69.8 | 77.3 | 82.8 | 85.0 | 86.6 | 90.4 | 90.9 | 92.0 | 93.3 |
| ATTENTIVEWALK | 64.2 | 76.7 | 81.0 | 83.0 | 87.1 | 88.2 | 91.4 | 92.4 | 93.0 |
| TADW | 86.6 | 88.2 | 90.2 | 90.8 | 90.0 | 93.0 | 91.0 | 93.0 | 92.7 |
| TRIDNR | 85.9 | 88.6 | 90.5 | 91.2 | 91.3 | 92.4 | 93.0 | 93.6 | 93.7 |
| CENE | 72.1 | 86.5 | 84.6 | 88.1 | 89.4 | 89.2 | 93.9 | 95.0 | 95.9 |
| CANE | 86.8 | 91.5 | 92.2 | 93.9 | 94.6 | 94.9 | 95.6 | 96.6 | 97.7 |
| DMTE | 91.3 | 93.1 | 93.7 | 95.0 | 96.0 | 97.1 | 97.4 | **98.2** | **98.8** |
| GAP | **95.8** | **96.4** | **97.1** | **97.6** | **97.6** | **97.6** | **97.8** | 98.0 | 98.2 |
| GAIN% | 4.5% | 3.6% | 3.4% | 2.6% | 1.6% | 0.5% | 0.4% | | |

Table 3: AUC score for link prediction on the Cora dataset

| Algorithm | % of training edges | | | | | | | | |
|---|---|---|---|---|---|---|---|---|---|
| | 15% | 25% | 35% | 45% | 55% | 65% | 75% | 85% | 95% |
| DEEPWALK | 56.6 | 58.1 | 60.1 | 60.0 | 61.8 | 61.9 | 63.3 | 63.7 | 67.8 |
| LINE | 52.3 | 55.9 | 59.9 | 60.9 | 64.3 | 66.0 | 67.7 | 69.3 | 71.1 |
| NODE2VEC | 54.2 | 57.1 | 57.3 | 58.3 | 58.7 | 62.5 | 66.2 | 67.6 | 68.5 |
| WALKLETS | 50.7 | 51.7 | 52.6 | 54.2 | 55.5 | 57.0 | 57.9 | 58.2 | 58.1 |
| ATTENTIVEWALK | 69.4 | 68.0 | 74.0 | 75.9 | 76.4 | 74.5 | 74.7 | 71.7 | 66.8 |
| TADW | 52.3 | 54.2 | 55.6 | 57.3 | 60.8 | 62.4 | 65.2 | 63.8 | 69.0 |
| TRIDNR | 53.8 | 55.7 | 57.9 | 59.5 | 63.0 | 64.2 | 66.0 | 67.5 | 70.3 |
| CENE | 56.2 | 57.4 | 60.3 | 63.0 | 66.3 | 66.0 | 70.2 | 69.8 | 73.8 |
| CANE | 56.8 | 59.3 | 62.9 | 64.5 | 68.9 | 70.4 | 71.4 | 73.6 | 75.4 |
| DMTE | 58.4 | 63.2 | 67.5 | 71.6 | 74.0 | 76.7 | 78.7 | 80.3 | 82.2 |
| GAP | **72.6** | **77.9** | **81.2** | **80.8** | **81.4** | **81.8** | **82.0** | **83.7** | **86.3** |
| GAIN% | 3.2% | 9.9% | 7.2% | 5.1% | 5.0% | 5.1% | 3.3% | 3.4% | 4.1% |

Table 4: AUC score for link prediction on the Zhihu dataset

| Algorithm | % of training edges | | | | | | | | |
|---|---|---|---|---|---|---|---|---|---|
| | 15% | 25% | 35% | 45% | 55% | 65% | 75% | 85% | 95% |
| DEEPWALK | 69.2 | 71.4 | 74.1 | 74.7 | 76.6 | 76.1 | 78.7 | 75.7 | 79.0 |
| LINE | 65.6 | 71.5 | 73.8 | 76.0 | 76.7 | 77.8 | 78.5 | 77.9 | 78.8 |
| NODE2VEC | 66.4 | 68.6 | 71.2 | 71.7 | 72.7 | 74.0 | 74.5 | 74.4 | 76.1 |
| WALKLETS | 70.3 | 73.2 | 75.2 | 78.7 | 78.2 | 78.1 | 78.9 | 80.0 | 78.5 |
| ATTENTIVEWALK | 68.8 | 72.5 | 73.5 | 75.2 | 74.1 | 74.9 | 73.0 | 70.3 | 68.6 |
| GAP | **77.6** | **81.6** | **81.9** | **83.3** | **83.1** | **84.1** | **84.5** | **84.8** | **84.8** |
| GAIN% | 7.3% | 8.4% | 6.7% | 4.6% | 4.9% | 6.0% | 5.6% | 4.8% | 5.8% |

Table 5: AUC score for link prediction on the Email dataset

smaller values of percentage of edges used for training. This is shows that GAP is suitable both in cases where there are several missing links and most of the links are present.

## 4.2 NODE CLUSTERING

Nodes in a network has the tendency to form cohesive structures based on some kinds of shared aspects. These structures are usually referred to as groups, clusters or communities and identifying them is an important task in network analysis. In this section we use the Email dataset that has ground truth communities, and there are 42 of them. Since this dataset has only structural information, we have excluded the baselines that require textual information.

| Algorithm | % of training edges | | | | | | | |
|---|---|---|---|---|---|---|---|---|
| | **25%** | | **55%** | | **75%** | | **95%** | |
| | NMI | AMI | NMI | AMI | NMI | AMI | NMI | AMI |
| DEEPWALK | 41.3 | 28.6 | 53.6 | 44.8 | 50.6 | 42.4 | 57.6 | 49.9 |
| LINE | 44.0 | 30.3 | 49.9 | 38.2 | 53.3 | 42.6 | 56.3 | 46.5 |
| NODE2VEC | 46.6 | 35.3 | 45.9 | 35.3 | 47.8 | 38.5 | 53.8 | 45.5 |
| WALKLETS | 47.5 | 39.9 | 55.3 | 47.4 | 54.0 | 45.4 | 50.1 | 41.6 |
| ATTENTIVEWALK | 42.9 | 30.0 | 45.7 | 36.5 | 44.3 | 35.7 | 47.4 | 38.5 |
| GAP | **67.8** | **58.8** | **64.7** | **55.7** | **65.6** | **57.6** | **65.4** | **58.7** |
| **%Gain** | 20.3% | | 9.4% | | 11.0% | | 7.8% | |

Table 6: NMI and AMI scores for node clustering experiment on the Email dataset. The Gain is with respect to the NMI only.

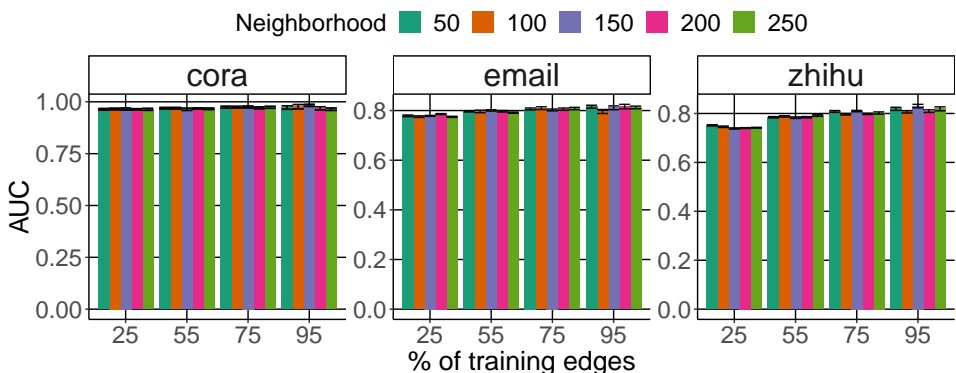

Figure 2: Sensitivity of GAP to the size of node's neighborhood $(S, T)$ on the link prediction task

**Setup:** Since each node belongs to exactly one cluster, we employ the Spectral Clustering algorithm to identify clusters. The learned representations of nodes by a certain algorithm are the input features of the clustering algorithm. In this experiment the percentage of training edges varies from 25% to 95% by a step of 20%, for the rest we use the same configuration as in the above experiment.

Given the ground truth community assignment $y$ of nodes and the predicted community assignments $\hat{y}$, usually the agreement between $y$ and $\hat{y}$ are measured using mutual information $I(y, \hat{y})$. However, $I$ is not bounded and difficult for comparing methods, hence we use two other variants of $I$ (Vinh et al., 2010). Which are, the normalized mutual information $NMI(y, \hat{y})$, which simply normalizes $I$ and adjusted mutual information $AMI(y, \hat{y})$, which adjusts or normalizes $I$ to random chances.

**Results:** The results of this experiment are reported in Table 6, and GAP significantly outperforms all the baselines by up to 20% with respect to AMI score. Consistent to our previous experiment GAP performs well in both extremes for the value of the percentage of the training edges. Similar improvements are achieved for AMI score.

### 4.3 PARAMETER SENSITIVITY ANALYSIS

Here we first show the sensitivity of the main hyper-parameter of GAP, which is the size of the neighborhood, #Neighborhood $= S = T$. Figures 2 and 3(A) show the effects of this parameter both on link prediction and node clustering tasks. In both cases we notice that GAP is not significantly affected by the change of values. We show the effect across different values of percentage and fixed (55%) of training edges for link prediction and node clustering tasks, respectively. Regardless of the percentage of training edges, we only see a small change of AUC (Fig 2), and NMI and AMI (Fig 3-a) across different values of #Neighborhood.

Next, we analyze the run time of training GAP and our goal is to show the benefit of removing the encoder of APN and we do this by comparing GAP against CANE, which employs the exact APN ar-

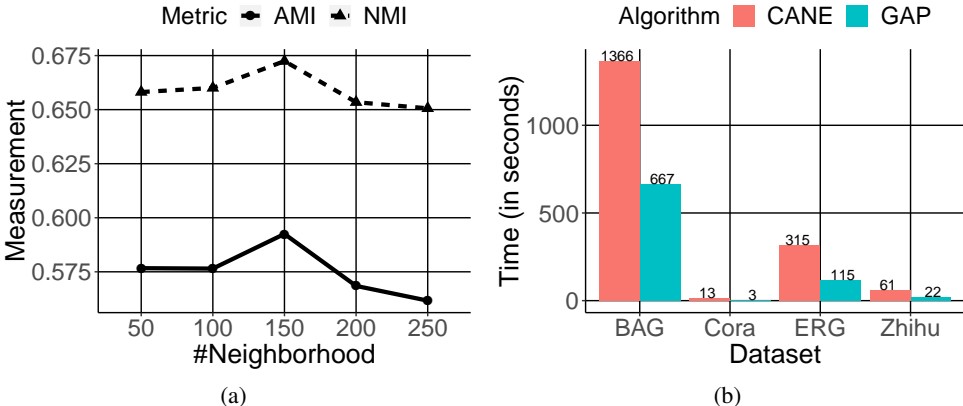

Figure 3: (a) Sensitivity of GAP to the size of node's neighborhood on the node clustering task and (b) the run time comparison of GAP and CANE using two real and two synthetic datasets

chitecture. For this experiment we include two randomly generated graphs, using ErdősRényi (ERG) and BarabásiAlbert (BAG) models. ERG has 200K edges, and BAG has 1.5M edges. Figure 3(B) clearly shows that GAP is at least 2 times faster than CANE in all the graphs.

## 5 RELATED WORK

NRL is usually carried out by exploring the structure of the graph and meta data, such as node attributes, attached to the graph (Perozzi et al., 2014; Grover & Leskovec, 2016; Tang et al., 2015; Perozzi et al., 2016; Wang et al., 2016; Yang et al., 2015; Pan et al., 2016; Sheikh et al., 2019). Random walks are widely used to explore local/global neighborhood structures, which are then fed into a learning algorithm. The learning is carried out in unsupervised manner by maximizing the likelihood of observing the neighbor nodes and/or attributes of a center node.

Recently graph convolutional networks have also been proposed for semi-supervised network analysis tasks (Kipf & Welling, 2017; Hamilton et al., 2017; Wu et al., 2019; Velickovic et al., 2017; Abu-El-Haija et al., 2019). These algorithms work by way of aggregating neighborhood features, with a down-stream objective based on partial labels of nodes, for example. All these methods are essentially different from our approach because they are context-free.

Context-sensitive learning is another paradigm to NRL that challenges the sufficiency of a single representation of a node for applications such as, link prediction, product recommendation, ranking. While some of these methods (Tu et al., 2017; Zhang et al., 2018) rely on textual information, others have also shown that a similar goal can be achieved using just the structure of the graph (Epasto & Perozzi, 2019). However, they require an extra step of persona decomposition that is based on microscopic level community detection algorithms to identify multiple contexts of a node. Unlike the first approaches our algorithm does not require extra textual information and with respect to the second ones our approach does not require any sort of community detection algorithm.

## 6 CONCLUSION

In this study we present a novel context-sensitive graph embedding algorithm called GAP. It consumes node neighborhood as input feature, which are constructed based on an important assumption that their ordering is arbitrary. To learn representations of nodes GAP employs attentive pooling networks (APN). By exploiting the above assumption, it makes an important simplification of APN and gains more than 2X speed up over another SOTA method, which employs the exact APN. Furthermore, GAP consistently outperforms all the baselines and achieves up to $\approx 9\%$ and $\approx 20\%$ improvement over the best performing ones on the link prediction and node clustering tasks, respectively. In future we will investigate how node attributes can be incorporated and provide a theoretical framework on the relation between the neighborhood sampling and topological properties.

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

## A  APPENDIX

Here we give a brief ablation experiment to strengthen the case for GAP. Concretely, we seek to compare GAP with different baselines to further motivate our choice of (i) the way we model nodes using their neighborhood, (ii) the assumption that order in this neighborhood is not important, and (iii) the choice of the APN algorithm without the CNN or BI-LSTM encoder. To this end, we introduce the following baselines and apart from the indicated difference everything will be the same as GAP.

| Datasets | Cora | | | | Email | | | |
|---|---|---|---|---|---|---|---|---|
| **Baselines** | **Training ratio** | | | | **Training ratio** | | | |
| | 25% | 55% | 75% | 95% | 25% | 55% | 75% | 95% |
| GAPCN | 60 | 60 | 61 | 61 | 74 | 74 | 78 | 79 |
| GAPAPN | 59 | 60 | 60 | 65 | 74 | 77 | 78 | 78 |
| GAPMLP | 56 | 63 | 66 | 73 | 72 | 77 | 78 | 77 |
| GAP | 96 | 97 | 97 | 98 | 81 | 83 | 84 | 84 |

Table 7: AUC results for the variants of GAP using the Cora and Email datasets

1. First we assume order is important in the neighborhood of nodes based on neighbors similarity with the current node. We use a topological feature to induce order, which is common neighbors. This baseline is referred to as GAPCN and uses the exact APN model to capture "order".

2. Second, we use the same input as in GAP, but nodes' neighborhood is now randomly permuted and fed multiple times (concretely 5 times), and the exact APN model is employed; this baseline is referred to as GAPAPN

3. Finally we replace GAP's Attend component with a standard feed-forward neural network that consumes the same input (Embedding matrices $\mathbf{S}$ and $\mathbf{T}$) and also has the same learning objective specified in Eq. 2; the baseline is referred to as GAPMLP.

In Table 7 we report the results of the ablation experiment. This sheds some light on the assumptions and the design choices of GAP. For the reported results, GAPCN and GAPAPN use a CNN Encoder. In both cases, they quickly over-fit the data, and we found out that we have to employ aggressive regularization using a dropout rate of 0.95. In addition, for GAPCN we have also observed that as we increase the kernel size to greater values than 1 (up to 5) the results keep getting worse, and hence, what we reported is the best one which is obtained using a kernel size of 1. For example, with a kernel size of 3 and training ratios of 25, 55, 75, and 95 percent the AUC scores respectively dropped to 54, 55, 56, and 58 percent on the Cora dataset, and 66, 69, 78 and 77 percent on the Email dataset. We conjecture that this is due the model's attempt to enforce high-level neighborhood patterns (eg. a combination of arbitrary neighbors) that are not intrinsically governing the underlying edge formation phenomena. Rather, what is important is to effectively pay attention to the presence of individual neighbors both in $N_s$ and $N_t$ regardless of their order. Apparently, training this model is at least twice slower than GAP as it is also illustrated in Section 4.3.

In the case of GAPAPN, though the variations in AUC are marginal with respect to the change in the kernel size, the training time of these model has increased almost by an order of magnitude. Finally we see that the mutual attention mechanism (Attend component) plays an important role by comparing the results between the GAPMLP and GAP.

