# OpenReview forum: "GRAPH NEIGHBORHOOD ATTENTIVE POOLING"
_ICLR.cc/2020/Conference — Reject_

### Official Review · AnonReviewer3 · 2019-10-11
**Official Blind Review #3**

**Rating:** 3

**Review:**

This paper proposed a GAP (graph neighborhood attentive pooling) method to solve problems of node clustering and link prediction. The proposed idea is mostly inspired by the attentive pooling network approach (APN) that has been widely used in the NLP domains; indeed the detailed steps of the proposed method is almost a ``mirror” application of APN from question-answer pair ranking in NLP to the graph domains, and the only difference is that rather than using an RNN or LSTM to deal with the temporal orders in sentences (as in APN), the proposed method simply collect the first-order neighbors of the source node and target node and concatenate them without considering their orders. For the rest part, the authors used a hard margin loss function which is also the same as in APN.

The proposed method is very similar to the APN method and so its novelty is limited. Another concern is that although graph nodes do not have temporal orders as in word sentences, their relative connections manifested through the edges are important topological information that should be captured in contextualize a node. Unfortunately the authors almost totally ignored this information. The ignorance of the node orders appear to me not an advantage but instead a limitation, though it makes the computation and implementation much easier.

The graph attention network (GAT) is a very related method but I see no comparison with it in node clustering tasks. Also many recent methods on graph convolutional networks are not incorporated for comparison.

Based on the concerns of novelty, lack of considering node topologies, and lack of comparison with strongly related methods, it is hard to recommend acceptance of this paper.


**Experience Assessment:**

I have published one or two papers in this area.

**Review Assessment: Checking Correctness Of Derivations And Theory:**

I assessed the sensibility of the derivations and theory.

**Review Assessment: Checking Correctness Of Experiments:**

I carefully checked the experiments.

**Review Assessment: Thoroughness In Paper Reading:**

I read the paper thoroughly.

---

> ### Author Response · Authors · 2019-11-14
> **Justification of our contribution**
>
> The authors’ would like to thank the reviewer for taking their time to read the paper and provide insightful and constructive feedback. It is our mistake that we haven’t clearly explain some points, now we have added more clarification in the updated version of the paper to address the reviewer’s concerns.
>
> However, we respectfully disagree with some of the reviewer’s comments as discussed below. First, we believe that there is a novelty in adopting a solution to the problem of context-sensitive NRL from a completely orthogonal problem of Q&A ranking, and in further simplifying the original algorithm so that it feets our problem and also scale. Moreover, the proposed insights can also be useful to other domains where the nature of the problem matches the one in NRL.
>
> Second, the assumption that the order of neighbors is not irrelevant is actually a natural assumption in network analysis. Often times one can effectively capture the essential properties of the graph by ignoring such order. Furthermore, this is a common assumption in a related line of research eg. [1, 2, 3]. In addition, we have added an experiment which validated our assumption. We kindly ask the reviewer to refer to the appendix in the modified version of the paper, particularly the variant called GAPCN.
>
> Third, we would like to point out that GAT is only loosely related work to our approach; while there is a superficial relation simply because both methods use attention mechanism nonetheless with a different intention and design. In addition, unlike our approach, GAT is a member of Graph Convolutional Networks (GCNs) eg.[ 4, 5, 6, 7].  Most importantly GAT is not context-sensitive whereas GAP is and the goal of this study is to demonstrate the benefit of context-free over context-sensitive approaches without using external metadata, such as textual features and labels, and this has dictated the choice of the baselines. This is actually the prevalent trend in existing SOTA context-sensitive methods that we are aware of (eg. CANE, DMTE, Splitter), among which we have included two of them (CANE, DMTE). As we have pointed out in the paper, the source code of Splitter was not publicly available at the time of submission. The aforementioned issues, we have tried to further emphasize and clarify in the updated version of the paper. In addition, most GCNs eg.[4, 5, 6, 7] are semi-supervised and inducting, which is not the case for GAP.
>
> We hope that our response and the added clarification in the paper have addressed your concerns.
>
>
> 1. Bryan Perozzi, Rami Al-Rfou, and Steven Skiena. 2014. DeepWalk: online learning of social representations. In Proceedings of the 20th ACM SIGKDD international conference on Knowledge discovery and data mining (KDD '14). ACM, New York, NY, USA, 701-710. DOI: https://doi.org/10.1145/2623330.2623732
> 2.  Aditya Grover and Jure Leskovec. 2016. node2vec: Scalable Feature Learning for Networks. In Proceedings of the 22nd ACM SIGKDD International Conference on Knowledge Discovery and Data Mining (KDD '16). ACM, New York, NY, USA, 855-864. DOI: https://doi.org/10.1145/2939672.2939754
> 3. Ying, Rex et al. “Graph Convolutional Neural Networks for Web-Scale Recommender Systems.” Proceedings of the 24th ACM SIGKDD International Conference on Knowledge Discovery & Data Mining - KDD  ’18 (2018): n. pag. Crossref. Web.
> 4. Thomas N. Kipf, Max Welling, Semi-Supervised Classification with Graph Convolutional Networks (ICLR 2017)
> 5. Petar Veličković, Guillem Cucurull, Arantxa Casanova, Adriana Romero, Pietro Liò and Yoshua Bengio. Graph attention Networks. (ICLR 2018)
> 6. Sami  Abu-El-Haija,  Bryan  Perozzi,  Amol  Kapoor,  Hrayr  Harutyunyan,  Nazanin  Alipourfard,Kristina Lerman,  Greg Ver Steeg,  and Aram Galstyan.   Mixhop:  Higher-order graph convolu-tional architectures  via sparsified neighborhood  mixing.CoRR, abs/1905.00067,  2019.   URLhttp://arxiv.org/abs/1905.00067.
> 7. Felix  Wu,  Tianyi  Zhang,  Amauri  H.  Souza  Jr.,  Christopher  Fifty,  Tao  Yu,  and  Kilian  Q.  Wein-berger.  Simplifying graph convolutional networks.CoRR, abs/1902.07153, 2019.  URLhttp://arxiv.org/abs/1902.07153

---

### Official Review · AnonReviewer2 · 2019-10-21
**Official Blind Review #2**

**Rating:** 3

**Review:**

- Minimal theoretical novelty: The paper is too focussed on the empirical advantage achieved on the datasets used in the experiments.

- Regarding equation (2), what is the guarantee this modification depicted by the hard-margin loss will always lead to improved performance like what happened with the experimented datasets? or more practically what are the required conditions for it to perform well?

- Writing really needs to improve. There are too many typos and grammatical mistakes.
Examples include:
 -- p1: "to learn representation of graphs".
 -- p1: "on other contexts its coupled with".
 -- p1: "NRL studies have shown a context-sensitive approach significantly outperform previous context-free SOTA methods in link-prediction task."
 -- p3: "a more sophisticated neighborhood functions".
 -- p5: "reporeted"
 -- p7: "two other variant"
 -- p8: "and gain"

- The latter issue makes is sometimes tricky to follow the ideas presented.
Example:
 -- last two lines in Section 3.

- Last paragraph in Section 1 is pretty informative about the pros and cons of the method. It also rather admits the first issue mentioned here in the review.

Minor:
- p4: "Eg. " --> eg.

**Experience Assessment:**

I have read many papers in this area.

**Review Assessment: Checking Correctness Of Derivations And Theory:**

I carefully checked the derivations and theory.

**Review Assessment: Checking Correctness Of Experiments:**

I carefully checked the experiments.

**Review Assessment: Thoroughness In Paper Reading:**

I read the paper thoroughly.

---

> ### Author Response · Authors · 2019-11-14
> **Response, clarifications, and corrections.**
>
> The authors’ would like to thank the reviewers insightful comment on both technical and presentation matters. We would like to apologize for the typos, other presentation mistakes and that we have not clearly explained some notions. We have corrected the errors pointed out, have done a thorough check to avoid any related mistakes as much as possible, and added more clarification.
>
> What is guaranteed is that this objective is a proxy to maximize the likelihood of the graph, $\mathcal{L}(E)$,  which is the de facto standard in NRL. It leads to latent representations of nodes that preserve the properties of the graph. In addition, hard margin loss is not the only alternative, for instance we have achieved similar results by materializing $\mathcal{L}(E)$ using the log loss based on negative sampling [1]
>
> $$\mathcal{L}(E) = - \sum_{e \in E } \log P(e)$$
> $$ P(e = <s, t>)  = p(r_t | r_s) $$
>
> Therefore, hard margin is just a choice  for $\mathcal{L}(E)$ that worked for us and the improvement is not mainly because of it, as almost all the baselines have a similar formulation. Rather, our argument is that the performance improvement over the baselines is achieved (i) because of the way we model nodes using their neighborhood, (ii) by the design choice that lets the representations of a pair of nodes to be a result of mutual influence based on their neighborhood (what we call context-sensitive approach) so as to capture complex edge formation patterns. To further corroborate our claim, we have included an experiment using three variants of GAP with the same objective and showed GAP outperforms them. We kindly refer the reviewer to the Appendix in the updated version of the paper.
>
> While our work has much more focus on empirical than theoretical aspects it nevertheless complies with the ICLR call and we believe it would be very interesting to the audience of ICLR
>
> We hope that our response and the added clarification in the paper have addressed your concerns.
>
> [1] Tomas Mikolov, Ilya Sutskever, Kai Chen, Greg Corrado, and Jeffrey Dean. 2013. Distributed representations of words and phrases and their compositionality. In Proceedings of the 26th International Conference on Neural Information Processing Systems - Volume 2 (NIPS'13), C. J. C. Burges, L. Bottou, M. Welling, Z. Ghahramani, and K. Q. Weinberger (Eds.), Vol. 2. Curran Associates Inc., USA, 3111-3119.

---

### Official Review · AnonReviewer1 · 2019-10-23
**Official Blind Review #1**

**Rating:** 3

**Review:**

Summary:
The proposed paper adapts Attentive Pooling Network (ATN) for graphs, by noting that the order of
neighbors of any node in a graph does not matter unlike the neighboring words in a sentence (for
which APN was developed). A simple modification of removing the layer which encodes higher order
sequential properties in APN, eg n-gram like statistics, the APN is adapted to GAP. This allows
encoding context from the neighboring nodes for a specific pair to be compared. Applications on Link
Prediction and Node Clustering are demonstrated on three benchmark datasets.

Detailed comments:
- The paper is simple and easy to read and the modification of APN to GAP is also small while being
well motivated and empirically important.

- There are some confusions in the writing: p4 first paragraph says that "In principle one can learn
using all pairs of nodes, however that is not scalable, and hence we restrict learning between pairs
in E." However from and around Eq2 it appears that r_s.r_t^- is also computed, where (s,r^-) is not
an edge.

- Also the statement below Eq2 "The goal is to learn, in an unsupervised fashion..." is not correct
  specially for link prediction task, as having a training graph (with edges) is having annotations.

- Although, the removal of the `encoding' step from APN makes sense, it would also be possible that
  the encoding operation is learned to be an identity function automatically if the training data is
  presented appropriately. I would assume that learning in such a case might take longer, and might
  not achieve the performance achieved by GAP. An ablation experiment could be reported to support
  GAP further where encoder is kept in APN, but at the training same pair is presented multiple
  times with the order of the neighborhoods randomized.

- Similar to the experiment above, the order of the nodes could be arbitrarily fixed by some simple
  logic. Eg order the neighboring nodes sorted by the feature similarity with the current node. This
  can then be fed to APN and a baseline could be reported.

- I do not directly work with Link Prediction or Node Clustering tasks, so am not familiar with the
  literature and the baselines reported. However, it might be that many baselines do not use a
  discriminative objective. Perhaps just using the discriminative objective gives a lot of boost and
  using the context from neighbors is not very important? From Fig2 and Fig3a it seems that
  neighborhood size does not make a big difference? This should also be ablated. One experiment
  could be to use just the current feature and have a small MLP on it, and learn using the proposed
  discriminative objective. If any other experiment can be designed in similar lines, it should be
  included as well.

- Why are run time comparison given on simulated data and not real data?

- The experimental setup for training edge selection should be detailed more. In particular, I
  would recommend that true generalization in terms of nodes and edges should be ensured, by having
  no test node (i) appear in the train set and (ii) has an edge which connects it to a training
  node. If this is not used then the alternative scheme used should be explained and argued for.


I am not a direct expert in the area and felt that the paper was somewhat lacking. I am putting my initial rating
to the conservative side. I am very open to revising the rating based on the author responses and the other
reviewers' comments.

**Experience Assessment:**

I have read many papers in this area.

**Review Assessment: Checking Correctness Of Derivations And Theory:**

I assessed the sensibility of the derivations and theory.

**Review Assessment: Checking Correctness Of Experiments:**

I carefully checked the experiments.

**Review Assessment: Thoroughness In Paper Reading:**

I read the paper at least twice and used my best judgement in assessing the paper.

---

> ### Author Response · Authors · 2019-11-14
> **Clarification on some confusions we have caused and added experiments**
>
> The authors’ would like to thank the reviewer for taking their time to read the paper and provide insightful and constructive feedback, which have helped us to improved the quality of our work. In the following we put our responses and the additional tasks performed to address the reviewer’s concerns.
>
> - The number of negative samples is usually a small constant, 5, and it is based on this empirical fact that we have ignored the impact of the negative example, rs-. The training is still restricted to the actual edges and a constant number of negative examples per edge, which makes the cost still proportional to the number of edges,O(cE)=O(E) as c is a constant.
>
> - The statement “The goal is to learn, in an unsupervised fashion …” is to point out the fact that there are no extra labels to guide the training, other than maximizing the graph (edges) likelihood. All the baselines have a similar objective of maximizing the graph likelihood i.e., all baselines are annotated if edges are considered as annotations. Furthermore we would like to emphasize that we deal with node clustering task, which doesn’t directly benefit from such annotations.
>
> - We have carried out the suggested experiments in your comment and added the results in the Appendix section, and our assumptions and design choices are validated.
>
> - As pointed out in an earlier response, almost all the baselines have a similar objective (maximizing the graph likelihood) and GAP has no special advantage as compared to the baselines.
>
> - The run time comparisons are performed on both real (Cora and Zhihu) and simulated datasets (BAG and ERG). The simulated datasets are included to see if graphs with different edge formation patterns have an effect on the training time in addition to the effect of their size, we have further clarified that in the paper.
>
> - GAP is carefully implemented not to include test nodes in the neighborhood of a node. However, since our algorithm is transductive, as are the baselines, it can only be trained on the current structure of the graph. (This can be verified from the source code that we have provided) in the gap_data.py file lines 50-52,
> # Ensures that no node from the test set is sampled in the neighbhorhood of any node
> # mask_nodes is a variable containing test nodes.
> msk = np.in1d(neighborhood_matrix, mask_nodes).reshape(neighborhood_matrix.shape)
> neighborhood_matrix[msk] = 0
>
> We hope that our responses have addressed your concern.

---

### Decision · Program_Chairs · 2019-12-19

**Decision:**

Reject

**Comment:**

All three reviewers are consistently negative on this paper. Thus a reject is recommended.